# Fibrillization Process of Human Amyloid-Beta Protein (1–40) under a Molecular Crowding Environment Mimicking the Interior of Living Cells Using Cell Debris

**DOI:** 10.3390/molecules28186555

**Published:** 2023-09-10

**Authors:** Mitsuhiro Hirai, Shigeki Arai, Hiroki Iwase

**Affiliations:** 1Graduate School of Science and Technology, Gunma University, 4-2 Aramaki, Maebashi 371-8510, Gunma, Japan; 2National Institute for Quantum and Radiological Science and Technology, Tokai 319-1106, Ibaraki, Japan; arai.shigeki@qst.go.jp; 3Comprehensive Research Organization for Science and Society (CROSS), Tokai 319-1106, Ibaraki, Japan; h_iwase@cross.or.jp

**Keywords:** human amyloid-beta protein, fibrillization, molecular crowding, intracellular environment, neutron scattering, selective deuteration, ab initio modeling

## Abstract

Molecular crowding environments play a crucial role in understanding the mechanisms of biological reactions. Inside living cells, a diverse array of molecules coexists within a volume fraction ranging from 10% to 30% *v*/*v*. However, conventional spectroscopic methods often face difficulties in selectively observing the structures of particular proteins or membranes within such molecularly crowded environments due to the presence of high background signals. Therefore, it is crucial to establish in vitro measurement conditions that closely resemble the intracellular environment. Meanwhile, the neutron scattering method offers a significant advantage in selectively observing target biological components, even within crowded environments. Recently, we have demonstrated a novel scattering method capable of selectively detecting the structures of targeted proteins or membranes in a closely mimicking intracellular milieu achieved utilizing whole-cell contents (deuterated-cell debris). This method relies on the inverse contrast matching technique in neutron scattering. By employing this method, we successfully observed the fibrillization process of human amyloid beta-protein (Aβ 1–40) under a molecular crowding environment (13.1% *w*/*v* cell debris, Aβ/cell debris = ~1/25 *w*/*w*) that closely mimics the interior of living cells. Aβ protein is well known as a major pathogenic component of Alzheimer’s disease. The present results combining model simulation analyses clearly show that the intracellular environment facilitates the potential formation of even more intricate higher-order aggregates of Aβ proteins than those previously reported.

## 1. Introduction

In living cells, large amounts of various types of molecules coexist in about 20–40% *v*/*v*, which is called a molecular crowding environment [1,2]. Therefore, to understand the biological reactions, it is crucial to elucidate the functional properties of molecules within living organisms, particularly within cells, where diverse molecules constitute the cellular environment. Numerous studies have been conducted to investigate the physicochemical properties of proteins, nucleic acids, lipids, and other molecules in the context of molecular crowding. Molecular crowding environments are generally considered to alter the equilibrium states of biological macromolecules via changes in chemical potential and activity, leading to the emergence of other equilibrium states [3,4]. In particular, how molecular crowding environments affect the structure, stability, function, and dynamics of proteins is essential for comprehending intracellular biological processes. The presence of large amounts of co-solutes (osmolytes) certainly alters the equilibrium states of proteins to those different in a non-crowding environment (diluted solution environment) and influences protein folding and stability [5,6]. However, the theoretical interpretation of the molecular crowding effect in living cells remains highly complex and controversial, and even extensive in silico approaches applied to intracellular environments face challenges in accurately predicting the effect on specific molecules [7,8].

On the other hand, the majority of experimental studies on biological molecules in solutions have typically been performed under dilute-solution conditions. Even previous studies involving molecular crowding environments were often carried out using simplified conditions with only a few simple components. Consequently, it is challenging to assert that these experimental conditions adequately reflect the true physicochemical environments of living cells and experimental approaches to observe the molecular structure and dynamics in intracellular environments remain few. One of the barriers in molecular crowding experiments, particularly when using spectroscopic methods, is the frequent encounter of serious difficulties in selectively observing the structure of a specific protein in the environment due to high backgrounds from crowder molecules. For such difficulties, neutron scattering, which can use different types of contrast variation methods [9,10,11], has significant advantages to overcome these difficulties in observing selectively certain biological components in crowded environments. Recently, we showed that the effect of the molecular crowding environment (sugars, polyol, denaturant, synthetic polymer) on the structure of a protein and its hydration as well as its stability against chemical and thermal perturbations, was observable and analyzable in detail. In those studies, we employed the inverse contrast matching method using deuterated-crowder molecules and/or the solvent contrast variation method. We demonstrated that small molecules, such as sugars and polyol [12,13,14], and a large molecule [15] affected protein structure and hydration in different ways depending on those concentrations and species. These differences were concluded to result from the variation in the activity coefficients of those molecules and water molecules that depends on molecular species, mass, and concentration [7,16,17]. However, unfortunately, the above experimental systems do not adequately consider the diversity of molecular species and molecular weights present in the crowded cellular environment. Therefore, it is crucial to develop in vitro measurement conditions that closely resemble the intracellular environment as much as possible to accurately observe the structural properties of specific proteins. We have recently reported a new in vitro measurement method that enables us to observe the structure of specific proteins or membranes in a cell-mimicking molecular crowding environment, which includes the entire contents of the cell (cell debris) [18]. We demonstrated that the structures of a specific small globular protein or a specified small unilamellar vesicle (SUV) can be quantitatively determined despite the presence of a highly concentrated cell-debris solution, up to a weight ratio of 1:60 for protein/crowder (cell debris) and 1:40 for SUV/crowder (cell debris). This technique is fundamentally based on the principle of the inverse contrast matching method of neutron scattering, allowing for the selective extraction of structural information about specific proteins or membranes from the strong background scattering generated by large amounts of cell fragments that mimic the cellular environment. In this study, we further utilized the above new method and successfully observed the fibrillization process of human amyloid beta-protein (Aβ 1–40) in a molecular crowding environment that closely mimics the interior of living cells, containing all cellular components. 

Human Aβ proteins, the target proteins of this study, denote peptides that are crucially involved in Alzheimer’s disease as the main component of the amyloid plaques found in the brains of people with Alzheimer’s disease (AD). Since the proposal of the amyloid hypothesis as a pathogenesis of AD [19], numerous studies have been conducted. The generation, aggregation, fibril formation, and accumulation of Aβ protein in nerve cells are the pathological features of AD. Aβ is cleaved from amyloid precursor protein and produced particularly through the process of proteolysis by β-secretase and γ-secretase [20]. Aβ aggregates, losing their solubility, form fibrillar structures and accumulate around nerve cells as amyloid plaques [21]. Such accumulation of Aβ within nerve cells disrupts important cellular processes such as mitochondrial function and calcium homeostasis, eventually leading to neuronal death [22]. As a result, it causes dysfunction in nerve cells and disrupts normal information transmission [23]. Not only Aβ fibrils, but Aβ oligomers during the aggregation stage of the Aβ protein are also suggested to have harmful effects on nerve cells [24]. In addition, several studies have directly observed the oligomerization, aggregation, and fibril formation of Aβ protein using various techniques such as X-ray scattering/diffraction, electron microscopy, and spectroscopy. For instance, X-ray diffraction analysis was used to reveal the polymorphism of Aβ protein fibrils in AD. Different forms and aggregation states of Aβ protein exhibited distinct diffraction patterns, suggesting that specific structural forms may be associated with the pathological features of AD [25]. Moreover, the investigation of Aβ protein’s fibrillar structure using a combination of electron microscopy and X-ray scattering showed the adoption of a cross-β structure, with certain structural units of Aβ fibrils suggested as being implicated in the development of AD pathology [26]. 

There are notable experimental and theoretical studies exploring the relationship between molecular crowding and the oligomerization, aggregation, and fibril formation of Aβ and other human amyloid proteins (e.g., α-synuclein). Experimental studies often use neutral polymers like polyethylene glycols, polysaccharides, or polyamines to investigate amyloid transition and fibril formation [27,28,29,30]. They showed that the molecular crowding environment can modulate the aggregation kinetics and fibril formation of Aβ, leading to the increased nucleation and fibril growth rates of Aβ protein. Theoretical studies simplified the shapes of crowding agents to spheres or cylindrical shapes, using their radius and volume fraction as parameters due to the essential role of the excluded volume effect in molecular crowding [8,31,32,33]. They suggested that both crowder volume fraction and crowder diameter have a large impact on fibril and oligomer formation. 

However, the molecular crowding conditions used in the above studies are considerably simplified and modeled special molecular crowding environments. Thus, those conditions do not adequately reflect the complexity of the actual intracellular environments. Therefore, we have developed a novel method harnessing the advantages of neutron scattering, which utilizes a molecular crowding environment closely mimicking the intracellular milieu containing all cellular contents. It should be noted that the molecular crowding environment used in this experiment is significantly different from what has been used in previous experimental and theoretical studies. This innovative method has enabled us to selectively observe and analyze the structure of Aβ protein aggregate in a molecular crowding environment where there are large amounts of intracellular components with complex compositions. The present results combining model simulations clearly demonstrate that the intracellular environment facilitates the potential formation of even more intricate higher-order aggregates of Aβ proteins than those previously reported.

## 2. Results and Discussion

### 2.1. Observation and Structural Characterization of Deuterated Cell Debris Solution: A Model System Closely Mimicking the Intracellular Molecular Crowding Environment

X-ray and neutron solution scattering techniques are highly convenient and effective experimental methods for in situ measurements of structures of biomolecules or synthetic polymers dissolved in solutions under various environmental conditions. Typically, these methods obtain the scattering curves of specific biomolecules or synthetic polymers of interest by subtracting the solvent scattering, which serves as the background, from the solution scattering that includes the target molecules. In other words, the structural information of solute particles in a solution is provided by the difference in the average scattering densities between the solute particles and the solvent, which is called ‘contrast’, Δ*ρ*. On the other hand, the major difference between the neutron scattering method and the X-ray scattering method is that the former utilizes particle waves with nuclear spin that interact with atomic nuclei, while the latter employs electromagnetic waves that interact with electrons. As a result, neutron scattering is known to have some prominent advantages compared to X-ray scattering without altering the physicochemical conditions of the solvent environment (such as pH or ionic strength), as shown below [10,11]. (1) The scattering length (nuclear scattering amplitude) of hydrogen and deuterium is comparable to that of other constituent elements (carbon, nitrogen, oxygen, phosphorus, etc.). (2) The average scattering densities of the main constituents of living organisms (proteins, nucleic acids, lipids, sugars, etc.) differ greatly from each other, and their values are located between the average scattering densities of light water and heavy water. Consequently, by simply varying the ratio of light water to heavy water in the solvent, it is possible to selectively observe the structures of each constituent component of the target molecule (called the ‘solvent contrast variation method’) [9]. Furthermore, as a sample can be prepared by selectively deuterating (deuterated labeling) a specific component constituting the solutes (or a specific site of the target molecule), various types of neutron contrast variation methods, such as label triangulation method, inverse contrast variation method, triple isomorphic replacement method, and nuclear spin contrast variation method, can be employed to selectively analyze the desired structure [10,11]. In addition, importantly, it is also important that in X-ray scattering, radiation damage to solute molecules caused by radicals generated by X-ray irradiation is unavoidable, while neutron scattering does not induce such damage. Overall, these features make the neutron scattering method a powerful and advantageous technique for structural analysis for biological materials compared to X-ray scattering. In neutron scattering measurements of biomolecules and/or synthetic polymers in solutions, the presence of light hydrogen (^1^H) in solutes and solvents leads to significant background scattering. This is because light hydrogen exhibits a much larger incoherent scattering cross section (~80 cm^−24^) compared to other elements (ex. ~2 cm^−24^ for ^2^H (D)). In other words, it is crucial to minimize the content of light hydrogen in solutes and solvents to obtain scattering data of the target molecule with high statistical accuracy. Therefore, to reduce incoherent background scattering as much as possible, it is important to use a solvent close to 100% D_2_O and employ deuterated compounds for solutes other than the target molecule. The aim of this study is to selectively observe the structural change and aggregation process of a specific protein (Aβ protein) in a molecular crowding environment close to in living cells. Hence, it is also necessary to mitigate the effects of incoherent scattering originating from the molecular crowding environment itself. To achieve such sample requirements, the cells were deuterated, and then their debris was dissolved in a solvent at a high concentration to mimic the cellular environment. 

Figure 1 shows the photograph of the cell debris solutions contained in the quartz sample cells with an optical path length of 1 mm used for the scattering measurements. Due to the presence of various types of molecules with different sizes in the solution, even at a concentration of 5%, it appears turbid. At 15%, it becomes almost opaque to visible light. Therefore, conventional spectroscopic measurements would become challenging.

Figure 2 shows the difference in scattering curves between the fully deuterated cell debris (cultured in deuterated cell growth medium) and non-deuterated cell debris (cultured in non-deuterated cell growth medium). The cell debris were dissolved in the D_2_O (*v*/*v*) solvent containing 1 mM EDTA, 10 mM HEPES at pD6.6 (pH 7.0). The concentration of the cell debris was 2% *w*/*v*. The ratio of the integrated scattering intensities in the *q* region below 1 Å^−1^ between the deuterated and non-deuterated cell debris is ~1/15, clearly demonstrating the effectiveness of deuterating the cells to reduce background scattering from the cell matrix for extracting the structure of specific molecules in a molecular crowding environment.

As we reported previously [18], the average neutron scattering densities of the cell debris cultured in deuterated and non-deuterated cell growth media are equivalent to those of 110% D_2_O and 33% D_2_O, respectively. To obtain the deuterated cell debris with an average scattering density close to 100% D_2_O, we used a mixture of deuterated and non-deuterated cell growth media in a ratio of 87/13 *v*/*v* for culturing the cells. Figure 3 shows the solution scattering curves of the deuterated-cell debris prepared from the cells cultured under the above condition. The cell debris was dissolved in aqueous solvents with different H_2_O/D_2_O ratios (0, 20, 40, 60, 80, 100% D_2_O (*v*/*v*)). The solvent condition and the cell debris concentration were the same as in Figure 1. In the high-*q* region of the scattering curves of the D_2_O solvent and the debris in D_2_O, the arrow at *q* = ~2 Å^−1^ corresponds to the correlation peak between water molecules. In the scattering curves of the solvents below *q* = ~1 Å^−1^ there is a tendency for the scattering intensity to remain unchanged, indicating the contribution of incoherent scattering. However, as the proportion of H_2_O in the solvent increases, the intensity rapidly increases. As evident from Figure 3, it is clear that as the ratio of D_2_O in the solvent decreases, the incoherent scattering significantly reduces. This indicates that using a solvent close to 100% D_2_O is crucial for obtaining the scattering from the solute with a high statistical accuracy.

Figure 4 presents the D_2_O/H_2_O ratio dependence of the deuterated-cell debris scattering curves obtained from the difference in the scattering curves of the solution and the solvent in Figure 3 according to the following formula. The scattering intensity of the solute (cell debris) was obtained using the following equation.
$$I_{solute}\left(q\right)=\left\{\frac{1}{B_{solution}T_{solution}}I_{solution}(q)-\frac{1}{B_{cell}T_{cell}}I_{cell}(q)\right\}-\left(1-cv_{a}\right)\left\{\frac{1}{B_{solvent}T_{solvent}}I_{solvent}(q)-\frac{1}{B_{cell}T_{cell}}I_{cell}(q)\right\}$$
where *I*_solute_(*q*), *I*_solution_(*q*), *I*_solvent_(*q*), and *I*_cell_(*q*) are the scattering intensities of the solute (cell debris), the debris solution, the solvent, and the sample cell, respectively; *B*_solution_, *B*_solvent_, and *B*_cell_ are the incident beam intensities; *T*_solution_, *T*_solvent_, and *T*_cell_ are the neutron transmissions, respectively. *c* and *v_a_* are the concentration of the solute and its partial specific volume. The *v_a_* value of cell debris can be roughly calculated from the constituents of organic compounds in the cell. It was approximately set to 0.74, estimated from the composition ratios of proteins, nucleic acids, lipids, carbohydrates, and other organic molecules, as well as inorganic ions in Escherichia coli, as reported in the scientific literature (e.g., those ratios in % *w*/*w* are approximately 15%, 7%, 2%, 3%, 1%, and 1%, with the remaining being water) [34]. In Figure 4, the scattering curve of the cell debris does not exhibit any specific peak or profile in spite of the presence of various kinds of molecules. When extracting structural information of a specific protein (here, non-deuterated Aβ protein) dispersed in a concentrated cell debris solution, it is crucial that the scattering curve from the cell debris does not exhibit distinct modulations, as the scattering from the cell debris becomes background scattering. In X-ray and neutron scattering methods, the plot of log *I*(*q*) vs. log *q* (the Porod plot) is useful when employed for characterizing the structural properties of objects [35,36]. The region in this plot where the scattering intensity gradually decreases and exhibits a linear slope is called the Porod region, which is primarily associated with the shapes of particles, the properties of interfaces, and the aggregate structures. There exists a specific relationship between the gradient of the scattering curve in the Porod region and the fractal dimension [37]. Fractal geometry is a natural description for disordered objects. The deuterated-cell debris used in this measurement is clearly composed of various types of biological components with different molecular weights, and is considered to be quite polydisperse. In Figure 4, the slope of the scattering curve in the Porod region is between −2.5 and −2, suggesting that the cell debris is characterized by the feature of a diffusion-limited aggregate and/or multi-particle diffusion-limited aggregate [38].

The square root of the zero-angle scattering intensity, *I*(0)^1/2^ obtained by extrapolating the observed scattering intensity to *q* = 0 is known to be proportional to (*cM*)^1/2^ Δ*ρ* [9,38], where *c*, *M*, and Δ*ρ* are the mass concentration, the average molecular weight (unknown in the present case), and the contrast of a solute particle, respectively. Here, instead of using *I*(0)^1/2^, we utilized the square root of the integrated scattering intensity, [*I*_integ_]^1/2^, in the smallest-*q* region from 0.0065 Å^−1^ to 0.009 Å^−1^. Figure 5 depicts [*I*_integ_]^1/2^ plotted against the D_2_O concentration. From the intercept value on the horizontal axis, the contrast-matching point of the deuterated-cell debris was determined to be 96.7 ± 0.3% D_2_O. Since the solvent with a 96.7% D_2_O was expected to minimize the contribution to the background scattering from the debris, we used this solvent in the subsequent measurements.

### 2.2. Time Evolution of the Process of Fibril Formation of Human Amyloid-Beta (1–40) under a Molecular Crowded Environment Using Cell Debris Closely Mimicking the Interior of Living Cells

After dissolving the Aβ protein in the 96.7% D_2_O solvent to a concentration of 4% *w*/*v*, the resulting Aβ solution was mixed in the volume ratio of 1:7 with either the 96.7% D_2_O solvent or the deuterated-cell debris solution (15% *w*/*v*, in 96.7% D_2_O). Subsequently, these mixed solutions were promptly used for the time-resolved measurements. The 96.7% D_2_O solvent condition was the same as in Figure 2. The final concentrations of Aβ protein and deuterated-cell debris in the respective solutions were 0.5% *w*/*v* Aβ protein and 0% *w*/*v* cell debris in the former solution, and 0.5% *w*/*v* Aβ protein and 13.1% *w*/*v* cell debris in the latter, respectively. In the latter solution, the weight ratio of Aβ protein to the cell debris is approximately 1:25 *w*/*w*. Figure 6A,B depict the time evolution of the scattering curves for the Aβ protein alone and the Aβ protein within the deuterated cell debris, respectively. The inset in Figure 6A serves as the control, showing that the deuterated cell debris, without the addition of Aβ protein, does not exhibit any changes. In Figure 6A,B, the broad humps of scattering intensity can be recognized, located at ~0.4 Å, indicated by the open arrows. These humps are attributable to the distance correlation between the β-pleated sheets, whose formation and stacking constitute the fundamental structure of amyloid aggregates leading to fibril formation, as shown in the following simulation analysis. In Figure 6A, the scattering intensity in the small-*q* region below 0.01 Å^−1^ gradually increased over 37 h, and the slope became steeper. The slope of the Porod plot (log *I(q*) vs. log *q*) becomes to change from −2.4 to −2.5, indicating the formation of the diffusion-limited aggregates [39]. However, no distinct peaks or profiles were observed. This indicates a progressive occurrence of heterogeneous aggregation or oligomerization of Aβ protein. Conversely, in the cell debris solution shown in Figure 6B, two prominent peaks at ~0.025 Å^−1^ and ~0.05 Å^−1^ started to emerge after 5 h of mixing. This clearly indicates that, instead of forming amorphous aggregates, a structural body with a certain periodicity was formed. These peaks indicate the presence of a periodic structure with a correlation distance of ~251 Å in real space. The intensities of these peaks increased over time. The matured amyloid fibril is known to have a diameter of approximately ~100–200 Å. Therefore, Figure 6B clearly indicates that the cell-debris environment, mimicking the interior of living cells, facilitated the maturation of amyloid fibrils within five hours and subsequently promoted the formation of even larger aggregates with a periodicity of ~251 Å.

### 2.3. Modeling Analysis of the Aggregate Structure of Amyloid Fibrils in Cell Debris Solution: Possibility of the Formation of Higher-Ordered Fibril

To explore periodic structure models that reproduce the observed scattering curve shown in Figure 6B, we conducted simulations of the scattering curves of fibril structures in solution using the atomic coordinates of amyloid β-peptide fibrils already registered in the Protein Data Bank (PDB). The PDB contains numerous atomic-resolution structures of cross-beta amyloid fibrils, obtained by combining experimental techniques such as NMR, X-ray fiber diffraction, cryo-electron microscopy, scanning transmission electron microscopy, and atomic force microscopy. Theoretical simulations of X-ray or neutron scattering curves for proteins in solution can be obtained using the CRYSOL (for X-ray) and CRYSON (for neutron) programs [39,40]. These programs are well-known for accurately reproducing experimental neutron scattering curves of proteins in aqueous solutions based on the atomic coordinates registered in the PDB, considering the hydration shell surrounding the protein surface. These programs are specifically designed to estimate and quantify structural and hydration changes in monodisperse solutions of known proteins with determined crystal structures. Therefore, there are limitations in directly applying them to the evaluation of the structures of polydisperse systems where amyloid aggregates grow over time. However, as an initial step, conducting simulations of theoretical scattering curves based on the extensively reported structures of amyloid aggregates and fibrils still hold some significance when qualitatively comparing them with experimentally observed scattering curves. Since we conducted the measurements in 96.7% D_2_O solvent, in the current simulation, we set the average scattering density of the solvent to be that of 96.7% D_2_O. The contrast of the hydration shell was set to 10% higher than the average scattering density of the solvent, which was the default value in the CRYSON program.

First, we examined how the difference in the symmetry of the fiber structure, specifically the variation in the beta-sheet structure, was reflected in the scattering curve. Figure 7 depicts a comparison of the theoretical neutron scattering curves for several types of assemblies of Aβ-peptide protofibril structures in 96.7% D_2_O with different fold symmetries. To facilitate the comparison of the characteristics of the scattering curves, the simulated theoretical scattering curves are normalized by the zero-angle scattering intensity and shown shifted. Here, we have selected several typical morphology models of Aβ amyloid fibrils that have already been reported. In Figure 7, 2m5n [41] and 2mxu [42] represent Aβ fibrils with double parallel-β-sheet and triple parallel-β-sheet segments, respectively. Additionally, 6ti5 [43], 5kk3 [44], and 2lmn [25] are Aβ fibrils exhibiting periodically twisted and two-fold symmetry, while 2lmq [25] represents an Aβ fibril with a three-fold symmetry. The dimensions and shapes of the structural units used for the present calculation, as illustrated in the 3D structures on the right-hand side of Figure 7, are nearly flat rectangular prisms (2m5n, 6ti5), elliptical cylinders (2mxu, 5kk3, 2lmn), and triangular prisms (2lmq), with heights along the fibril axis ranging from approximately 35 Å to 55 Å, respectively. In Figure 7, all scattering curves below *q* = ~0.1 Å^−1^, known as the small-angle scattering region, exhibit a saturation tendency in the scattering intensity. The scattering profiles in this region simply depend on the shape and dimensions of solute particles for homogeneous solutions. It should be mentioned that, in the case of heterogeneous solutions with a size distribution of solute particles, such as in the evolution of aggregation, the scattering intensity in the small-angle scattering region does not exhibit a saturation tendency but shows an increasing tendency. This is the present case, as shown in Figure 6A. In Figure 7, the difference in the slope of the scattering curve, especially in the *q*-region of ~0.1~0.2 Å^−1^, is mainly attributed to the variation in the shape of the structural units. As already demonstrated experimentally and theoretically in detail, the X-ray and neutron scattering curves of proteins in solution, covering scattering data in a wide *q*-range, contain the entire structural information of proteins at different hierarchical structural levels [45,46]. Specifically, the scattering curves in the regions of *q* < ~0.2 Å^−1^, ~0.25 Å^−1^ < *q* < ~0.5 Å^−1^, ~0.5 Å^−1^ < *q* < ~0.8 Å^−1^, and ~1.1 Å^−1^
*q* < ~1.9 Å^−1^ primarily correspond to distinct hierarchical structure levels, namely, the quaternary and tertiary structures, the inter-domain correlation and intra-domain structure, and the secondary structures including the closely packed side chains, respectively. Thus, the scattering curves in the *q*-region from ~0.25 Å^−1^ to ~0.8 Å^−1^ primarily reflect the geometrical packing arrangements of β-pleated sheets. The scattering curves of Aβ fibril models with two-fold or three-fold symmetry (6ti5, 5kk3, 2lmn) have a shoulder and/or hump at ~0.2 Å^−1^ due to the presence of inter-subunit correlation. Such a shoulder is not clearly seen in Figure 6. On the other hand, most of the model structures, except for 2mxu, exhibit distinct peaks or bumps around *q* = ~0.3 to 0.5 Å^−1^. This is mainly attributed to the distance correlation between the first and second adjacent parallel β-sheets. Similar peaks were observed experimentally, as shown in Figure 6.

Next, we examined how the growth of amyloid fibrils, specifically the elongation of the fibril structure, affects the scattering curve. Using the PyMOL program (http://www.pymol.org (accessed on 1 December 2015)), elongated fibril structures were generated by aligning the 2m5n model structure (unit structure) along its fiber axis. Figure 8 depicts the scattering curves of the elongated fibril models, with fibril lengths of approximately 35 Å for model-1, ~525 Å for model-2, ~1050 Å for model-3, and ~2100 Å for model-4, respectively. These model structures were constructed to have a pseudo-helical arrangement, as shown in the 3D structures on the right-hand side of Figure 8. As the fibril length increases, the scattering intensity region showing the Porod slope of −1 extends to a smaller *q*-region, consistent with the Porod slope of thin rod-shaped particles being −1. The *q*-region where the scattering intensity shows a saturation tendency also shifts to a smaller *q*-value range.

Through the integration of diverse techniques such as magic angle spinning nuclear magnetic resonance spectroscopy, X-ray fiber diffraction, cryo-electron microscopy, scanning transmission electron microscopy, and atomic force microscopy, alternative potential structures for the hierarchical assemblies of Aβ fibrils have been elucidated [41] and documented within the Protein Data Bank (PDB). PDB IDs 2m5n, 2m5k, 2m5m, and 3zpk correspond to the protofibril, dimeric, trimeric, and tetrameric assemblies of Aβ protofibrils, respectively. The higher-order packing arrangements of these models were constructed based on atomic-resolution data of a paired β-sheet protofilament structure [47]. These protofilaments are organized into multiple parallel strands perpendicular to the fiber axis, forming a multi-stranded helical ribbon structure. Utilizing the atomic coordinates provided in the PDB for the aforementioned dimeric, trimeric, and tetrameric assemblies, the theoretical neutron scattering curves of multiple helix arrangements of amyloid fibrils in 96.7% D_2_O were calculated. The 3D structures and paired β-sheet arrangements of the 2m5k, 2m5m, and 3zpk models are depicted on the right-hand side of Figure 7C. The dimensions of the unit structures are ~520 Å in fibril length for all cases, and 84 Å for 2m5k, 121 Å for 2m5m, and 154 Å for 3zpk in fibril width, respectively. The distance between paired β-sheets is around 25Å, while the distance between protofibrils is approximately 38Å. These correlation distances appear as a ripple feature of the scattering curves in the *q*-range from ~0.15 Å^−1^ to ~0.25 Å^−1^. In Figure 7C, the scattering curve of the 2m5n protofibril with a length of 525 Å, as previously depicted in Figure 6B, is presented once again. In Figure 9, despite assuming a higher-order amyloid fibril structure compared to Figure 7 and Figure 8, it is evident that reproduction of the correlation peaks at ~0.025 Å^−1^ and ~0.05 Å^−1^ observed experimentally in Figure 6B is not achievable.

To explore the formation of even higher-order aggregates, a super-structural arrangement was envisaged, considering the structures in Figure 9 as the fundamental units and postulating that they form higher-order assemblies withholding a certain distance between them. Figure 10A shows an example of the theoretical scattering curves when a pair of 3D models of 2m5k is arranged in parallel at fixed distances (225 Å, 250 Å, 275 Å), forming a dimer. In Figure 10A, the observed scattering curve showing correlation peaks at ~0.025 Å^−1^ and ~0.05 Å^−1^ (the scatter curve of Aβ protein aggregate formed after 37 h of dispersing Aβ proteins in cell debris solution) was also overlaid. While the position of the correlation peaks around 0.025 Å^−1^ and 0.05 Å^−1^ shift towards the small-*q* region as the distance between structural units increases, the observed correlation peaks can be qualitatively reproduced. The analysis mentioned above was carried out assuming possible structures for amyloid β-protein aggregates. Additionally, an alternative attempt was made to construct model structures without relying on such assumptions. We employed an ab initio modeling technique known as bead-modeling, utilizing the DAMMIF program [48]. This program begins with an arbitrary initial model and incorporates simulated annealing to create a compact interconnected model that generates a scattering pattern aligning with the experimental data. Figure 10B depicts the comparison between the theoretical scattering curve optimized using the ab initio modeling method and the experimental scattering curve of the Aβ protein observed 32.6 h after being mixed into the cell debris solution. The radius and number of the dummy atom (bead) were 25.5 Å and 1645, respectively. When executing the DAMMIF program, we did not assume that solute molecules adopt any specific shape or compact structure. Nonetheless, all the recommended model structures, which have discrepancies (normalized spatial discrepancy, NSD) between the model scattering function and experimental data below a certain threshold (NSD < 1.7), exhibit ripple features in the scattering curve at approximately 0.025 Å^−1^ and ~0.05 Å^−1^. This provides strong evidence for the formation of extended fibrous structures that spread while maintaining an appropriate spacing. The 3D model structure is illustrated on the right-hand side of Figure 10B.

Thus, the distinct modeling analyses described above are able to qualitatively explain the experimental data, suggesting the potential formation of even more intricate higher-order aggregates of Aβ proteins than those reported previously.

## 3. Materials and Methods

To obtain a crowder containing whole-cell contents with an appropriate deuteration ratio, *E. coli* (strain BL21) was cultured in a mixture of the deuterated and non-deuterated cell growth media (CGM-1000-D and CGM-1000-U; Cambridge Isotope Laboratories. Inc., Tewksbury, MA, USA). The ratio of the deuterated medium and the non-deuterated medium was 86/14 (*v*/*v*). As already reported, this ratio was chosen so that the average scattering density of the cultured *E. coli* cells did not exceed that of the D_2_O solvent. The growth medium was removed from the cultured cells by the treatment of lysis to pure solvent, stirring, and centrifuging several times. After this treatment, the cells were crushed by ultrasonic irradiation on ice cooling using a high-power probe-type ultrasonicator (Model UH-50; Shimazu Co., Kyoto, Japan), lyophilized and stored as the deuterated-cell debris crowder. Amyloid β-peptide (Human, 1–40) purchased from Peptide Institute Inc. (Osaka, Japan) was used without further purification. The solvents were D_2_O/H_2_O mixture at pD 6.7 (pH 7.1) with 2 mM EDTA (ethylenediaminetetraacetic acid), 10 mM HEPES (*N*-(2-hydroxymethyl) piperazine-*N*′-(2-ethane-sulfonic acid)). Deuterium oxide (99.9 atom % D) was obtained from Sigma-Aldrich. All other chemicals used for the above sample preparations described above were of analytical grade. For the determination of the contrast matching point, the lyophilized powder of the deuterated-cell crowder was dissolved in the solvents with different D_2_O concentration. The concentrations of these solutions were 2% *w*/*v*. The Aβ solution was prepared by dissolving Aβ powder in the D_2_O/H_2_O (96.7% D_2_O) solvent with an Aβ concentration of ~4% *w*/*v*. This Aβ solution and the deuterated cell crowder solution (15% *w*/*v*) or the 96.7% solvent was mixed in the volume ratio of 1:7. These mixtures were served for the time-resolved neutron scattering measurements. The final concentrations of Aβ and crowder in the mixture samples were ~0.5% *w*/*v* and 13.1% *w*/*v*, respectively.

The neutron scattering measurements were carried out using a small- and wide-angle neutron scattering (SWANS) spectrometer (BL15 TAIKAN) at the pulsed-neutron source of the Materials and Life Science Experimental Facility (MLF) of the Japan Proton Accelerator Research Complex (J-PARC, Tokai, Japan). The detailed spec of the TAIKAN spectrometer was already reported [49]. The neutron wavelength used was 0.5–6.0 Å. The sample solutions were contained in the quartz cells with a 1 mm path length. The temperature of the samples was controlled at 25 °C using a water-bath circulator. The exposure time of each measurement was from 0.5 to 1 h. In the neutron scattering measurements, we utilized the inverse contrast-matching method [50]. Our aim was to deuterate the cell debris, adjusting its average scattering density to match or approach that of the D_2_O solvent, thereby maximizing the excess average scattering density, known as ‘contrast’, of the nondeuterated Aβ protein in comparison to the solvent containing the cell debris. The method we employed enables the selective observation of the structure of a non-deuterated target protein within an environment that closely mimics intracellular molecular crowding. The modeling methods using the programs CRYSON and DAMMIF were employed to analyze the features of the observed scattering curves. Detailed explanations of these programs can be found in the original papers [39,40,48] and on their website (https://www.embl-hamburg.de/biosaxs/software.html (accessed on 6 June 2020)). The former program is particularly useful for characterizing the observed neutron scattering curve of a protein in solution, comparing it with the neutron scattering curves of proteins already registered in the Protein Data Bank. The latter (ab initio modeling) is applicable for deducing potential models of the unknown structure of a target material based on the observed scattering curves.

## 4. Conclusions

In this study, we focused on the process of fibrillization and aggregation of amyloid-β proteins, which are considered to be the causative agents of Alzheimer’s disease, among various reactions occurring within the molecular-crowding intracellular environment. By realizing a closely mimicking intracellular environment using a cell debris solution containing whole cell components, we have successfully demonstrated that the intracellular environment accelerates the fibrillization of amyloid-β proteins and the formation of higher-ordered aggregates. Such a molecular crowding effect on amyloid formation is in agreement with previous reports [27,28,30,32]. Various types of amyloid aggregate structures have been already found. Common features of amyloid aggregate structures are that a part and/or the entire structure of a protein can transition into misfolded structures rich in β-sheets, leading to the growth of stable amyloid proto-fibrils through aggregation and the accumulation of aggregates [51]. The experimental data of amyloid-β proteins in concentrated cell debris solution suggest the formation of novel amyloid aggregate structures with periodicity and/or correlation lengths of ~250 Å. Hence, simulation approaches based on the model structures of multi-stranded helical fibrils already reported, as well as ab initio simulations without assuming a prior structure, were conducted. These simulations successfully replicated the qualitative characteristics of the observed data, suggesting the potential formation of even more intricate higher-order aggregates of Aβ proteins than those previously reported. 

Another noteworthy aspect of this research lies in the observation and analysis of human amyloid-β protein aggregation within a concentrated solution of cellular debris, which retains the major components of the intracellular milieu. Due to the difficulty of experimental observations, most of the previous investigations into the structural stability of proteins have been conducted in diluted solutions or under the presence of specific coexisting molecules, often deviating from the actual intracellular environment. Therefore, the present study would stand out for its approach that closely mimics the cellular conditions. The feasibility of the measurements and analyses carried out in this study stems from the experimental advantages of neutron scattering techniques, which make use of sample deuteration and contrast variation. Figure 11 schematically depicts the strategy of the novel technique presented in this research, namely, the approach for selectively observing the structure of a particular molecule of interest within an intracellular molecular crowding environment. Figure 11A depicts the case for measurements of specific molecule structures within a molecular crowding environment using conventional experimental techniques such as X-ray scattering and spectroscopy. The observable data contain diverse information arising from the structures of all constituent molecules and their interactions. Figure 11A,B correspond to the scenarios for neutron scattering measurements utilizing non-deuterated compounds solubilized in H_2_O solvent and D_2_O solvent, respectively. Hence, in both instances, a significant amount of structural information originating from coexisting molecules and their interactions, apart from the targeted molecule of interest, is present in the observed data. Such unnecessary substantial information results in significant background noise, making the selective observation and analysis of the structure of specific molecules that are present in low quantities extremely challenging. The method adopted in this study corresponds to Figure 11C. In this method, by adjusting the cellular deuteration level, it becomes possible to achieve measurement conditions where the average scattering density of the major cellular constituents is nearly equivalent to that of heavy water. This minimizes unnecessary structural information from cellular debris (coexisting substances), leading to conditions that substantially reduce background noise. As a result, this enables us not only to enhance the contrast for the targeted molecules (non-deuterated) but also to selectively observe and analyze the structure of targeted molecules present in minute quantities within the intracellular environment. The ultimate application of this method would be to observe the structural changes and dynamics of specific proteins involved in various events that occur inside living cells. For instance, if specific proteins can be directly introduced into deuterated cells using the microinjection method, which is frequently employed to generate transgenic animals, it may enable such observations. The present method is anticipated to offer a promising technique for investigating the structure and dynamics of biological macromolecules within intracellular molecular crowding environments, which play a crucial role in governing biological systems. 

## Figures and Tables

**Figure 1 molecules-28-06555-f001:**
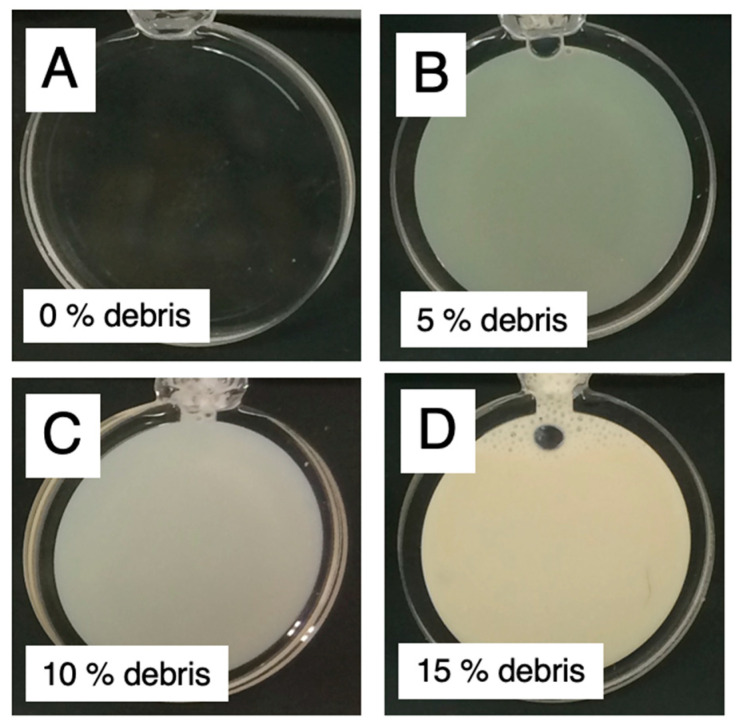
Photos of the cell debris solutions contained in the sample cell with an optical path length of 1 mm, consisting of a pair of quartz glass windows with a thickness of 0.5 mm each. These samples were used for neutron scattering measurements. (**A**), 0% *w*/*v* cell debris; (**B**), 5% *w*/*v* cell debris; (**C**), 10% *w*/*v* cell debris; (**D**), 15% *w*/*v* cell debris. The cell debris solution at high concentration was essentially nontransparent to visible light.

**Figure 2 molecules-28-06555-f002:**
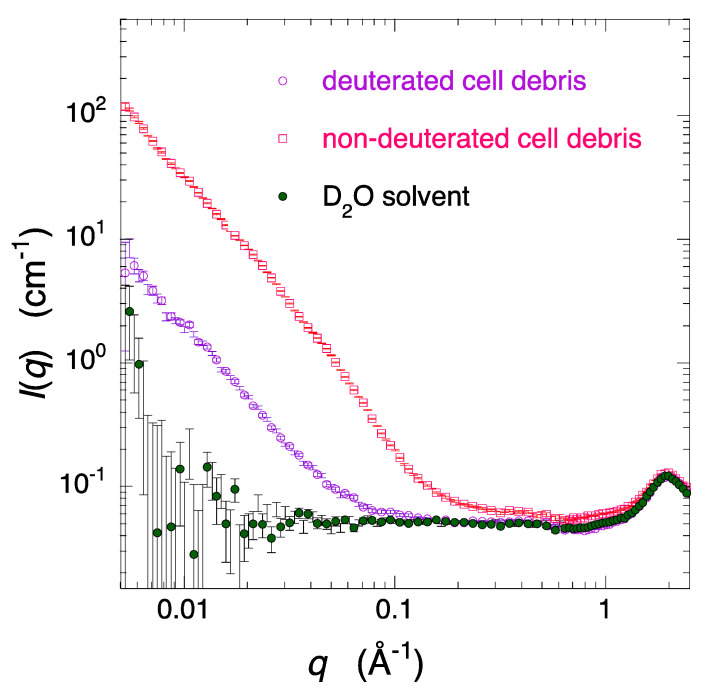
Comparison of scattering curves of the fully deuterated *E. coli* cell debris (cultured in deuterated cell growth medium) and the non-deuterated *E. coli* cell debris (cultured in non-deuterated cell growth medium) dissolved in 100% D_2_O (*v*/*v*) solvent (1 mM EDTA, 10 mM HEPES at pD6.6 (pH 7.0)). The concentration of the cell debris was 2% *w*/*v*. The absolute scattering intensity calibration was performed using H_2_O (in 1 mm path length at 25 °C).

**Figure 3 molecules-28-06555-f003:**
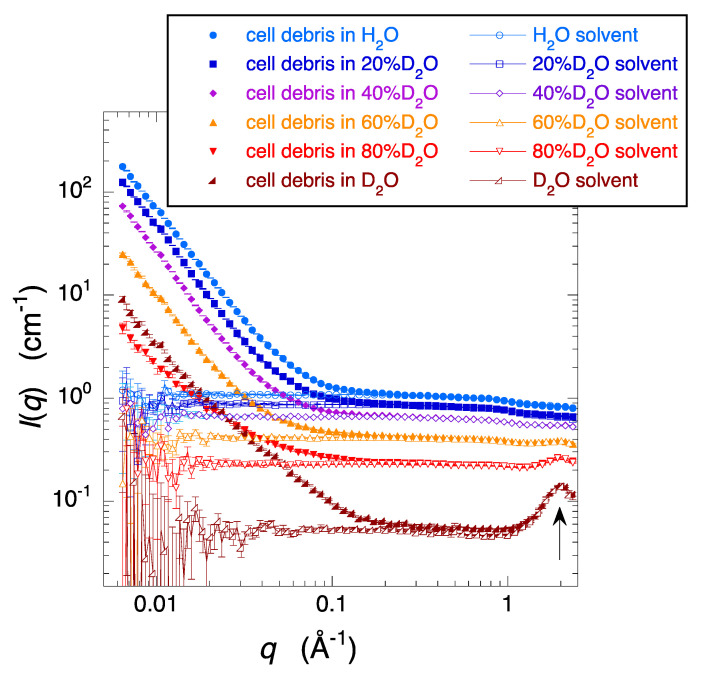
Scattering curves of the deuterated-cell debris (cultured by the mixture of deuterated and non-deuterated cell growth media in a ratio of 87/13 *v*/*v*) in the aqueous solvents with different D_2_O/H_2_O ratio (0, 20, 40, 60, 80, 100% *v*/*v* D_2_O) at 25 °C. The solvent condition and the cell debris concentration are the same as in Figure 2.

**Figure 4 molecules-28-06555-f004:**
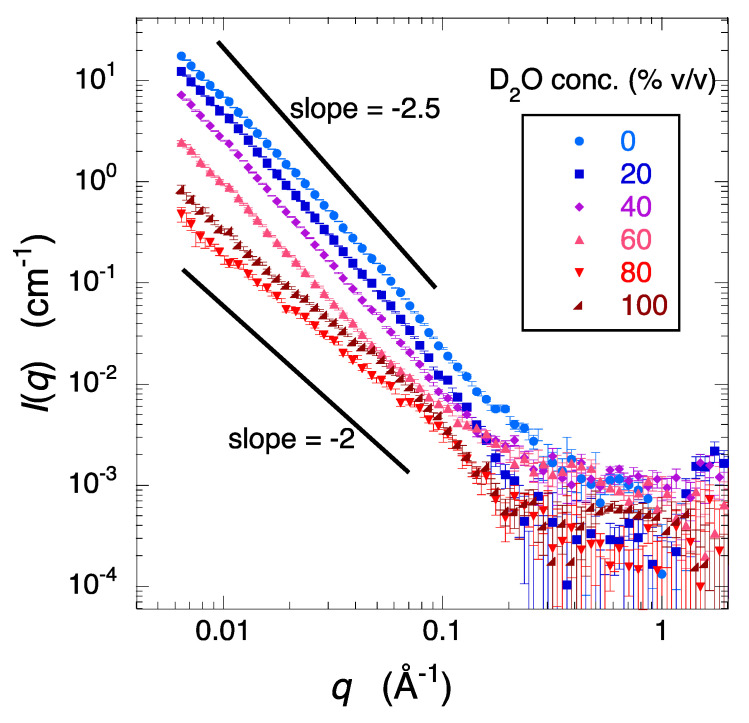
D_2_O/H_2_O ratio dependence of the deuterated-cell debris scattering curves obtained from the difference in the scattering curves of the solution and the solvent shown in Figure 3.

**Figure 5 molecules-28-06555-f005:**
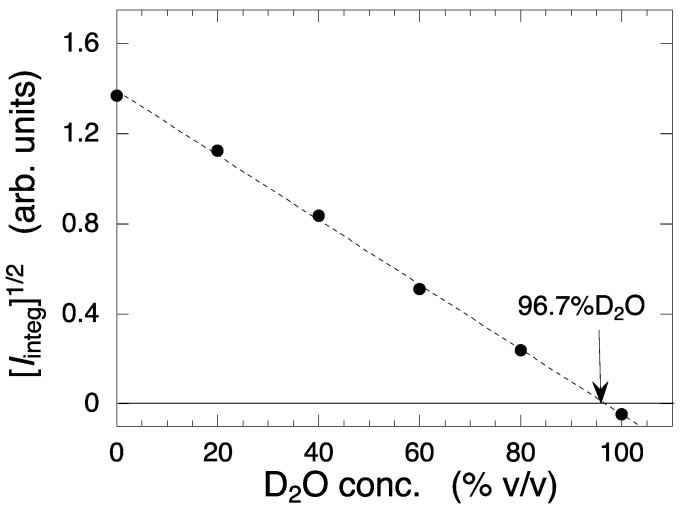
Square root of the integrated scattering intensity in the small-*q* region from 0.0065 to 0.009 Å^−1^ plotted against the D_2_O concentration. The integrated scattering intensities were obtained from Figure 4. The broken line is the linear regression line. From the intercept of the *x*-axis, the contrast matching point of the deuterated cell debris was determined as indicated by the arrow.

**Figure 6 molecules-28-06555-f006:**
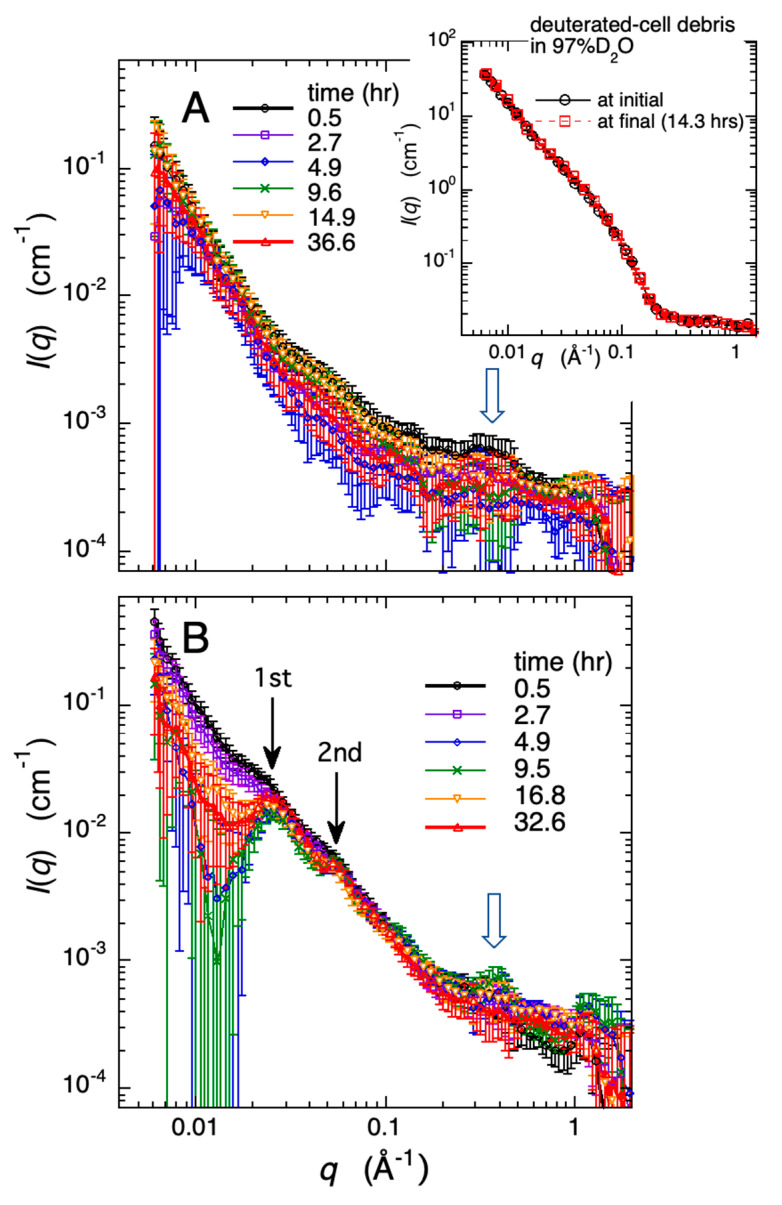
Time-evolution of the growth of amyloid fibril (human amyloid-β peptide 1–40) observed by wide-angle neutron scattering. In the figure, (**A**) represents 0.5% *w*/*v* Aβ protein in 96.7% D_2_O solvent, and (**B**) represents 0.5% *w*/*v* Aβ protein and 13.1% *w*/*v* deuterated-cell debris in 96.7% D_2_O solvent. The broad humps located at ~0.4 Å, indicated by the open arrows in (**A**,**B**), correspond to the distance correlation between inter-pleated β-sheets of Aβ fibrils. The scattering peaks located at ~0.025 Å^−1^ and ~0.05 Å^−1^, indicated by the solid arrows in (**B**), represent the 1st and 2nd order peaks associated with a structure having a periodicity of about 251 Å in real-space distance. In (**A**), the insert represents the control: the deuterated cell debris, without the addition of Aβ protein, does not exhibit any change.

**Figure 7 molecules-28-06555-f007:**
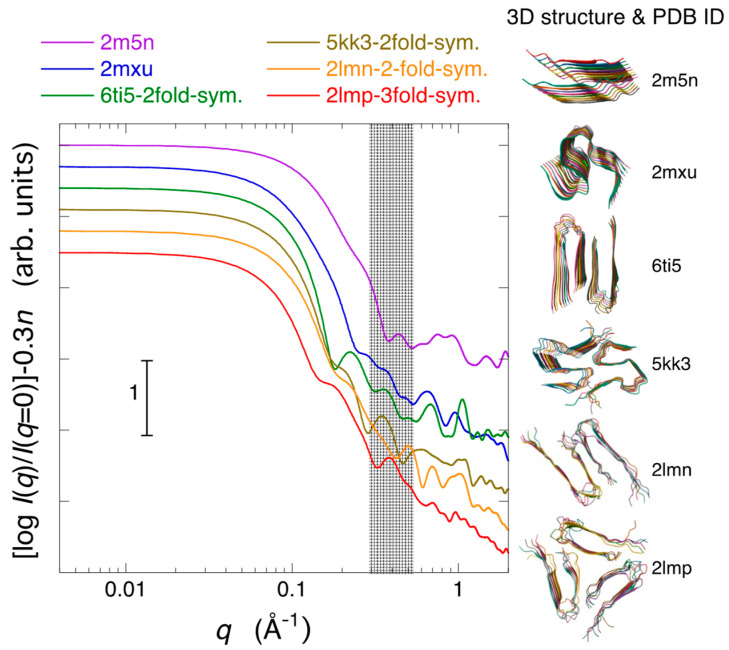
Comparison of theoretical neutron scattering curves for Aβ amyloid fibril models in 96.7% D_2_O with various fold symmetries of β-sheets. The atomic coordinates were obtained from the Protein Data Bank (PDB). Specifically, 2m5n represents a double parallel-β-sheet segment; 2mxu is a hinged triple parallel-β-sheet segment; 6ti5, 5kk3, and 2lmn correspond to Aβ fibrils displaying periodically twisted and two-fold symmetry; while 2lmq represents an Aβ fibril featuring three-fold symmetry. The three-dimensional structures are illustrated on the right-hand side.

**Figure 8 molecules-28-06555-f008:**
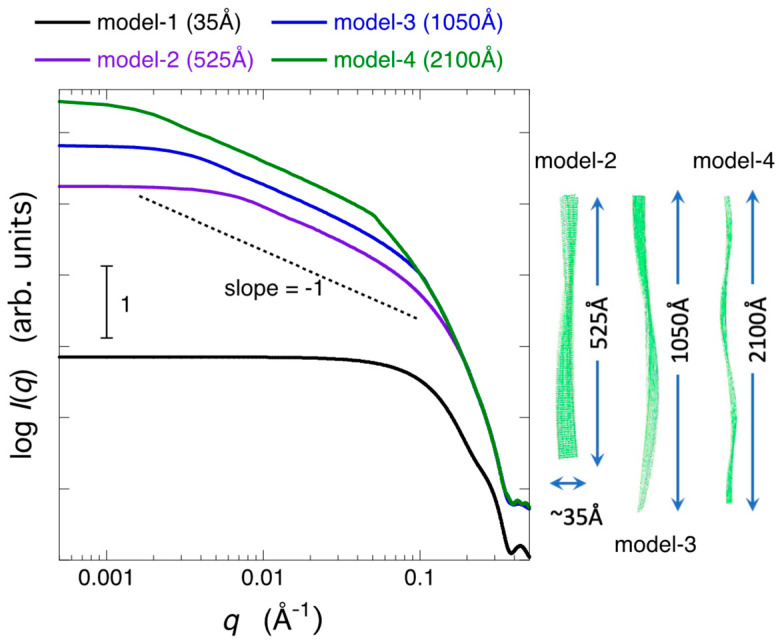
Protofibril length dependence of theoretical neutron scattering curves of the amyloid fibrils in 96.7% D_2_O solvent. The elongated fibril structures were generated by aligning the 2m5n model structure (unit structure) along its fiber axis. The fibril lengths are approximately 35 Å for model-1, ~525 Å for model-2, ~1050 Å for model-3, and ~2100 Å for model-4, respectively.

**Figure 9 molecules-28-06555-f009:**
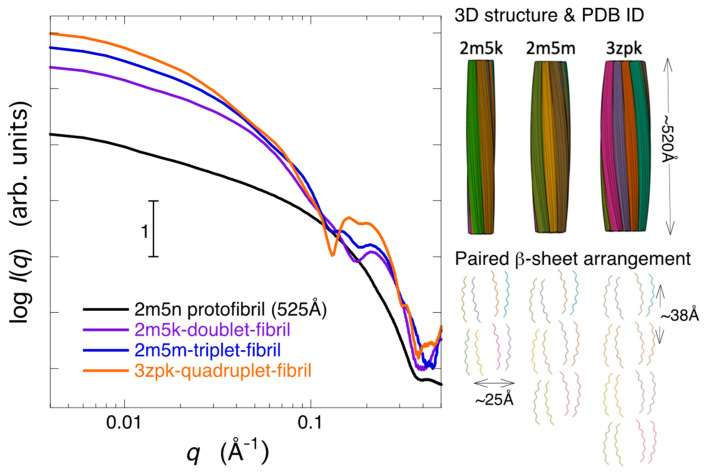
Comparison of the theoretical neutron scattering curves of multi-stranded helical amyloid fibrils in 96.7% D_2_O. PDB IDs 2m5k, 2m5m, and 3zpk correspond to doublet, triplet, and quadruplet amyloid fibrils, respectively. The unit structures have dimensions of approximately 520 Å in fibril length for all fibrils. The scattering curve for 2m5n, with a length of 525 Å, is the same as shown in Figure 8. The 3D structures and cross-sectional view are illustrated on the right-hand side.

**Figure 10 molecules-28-06555-f010:**
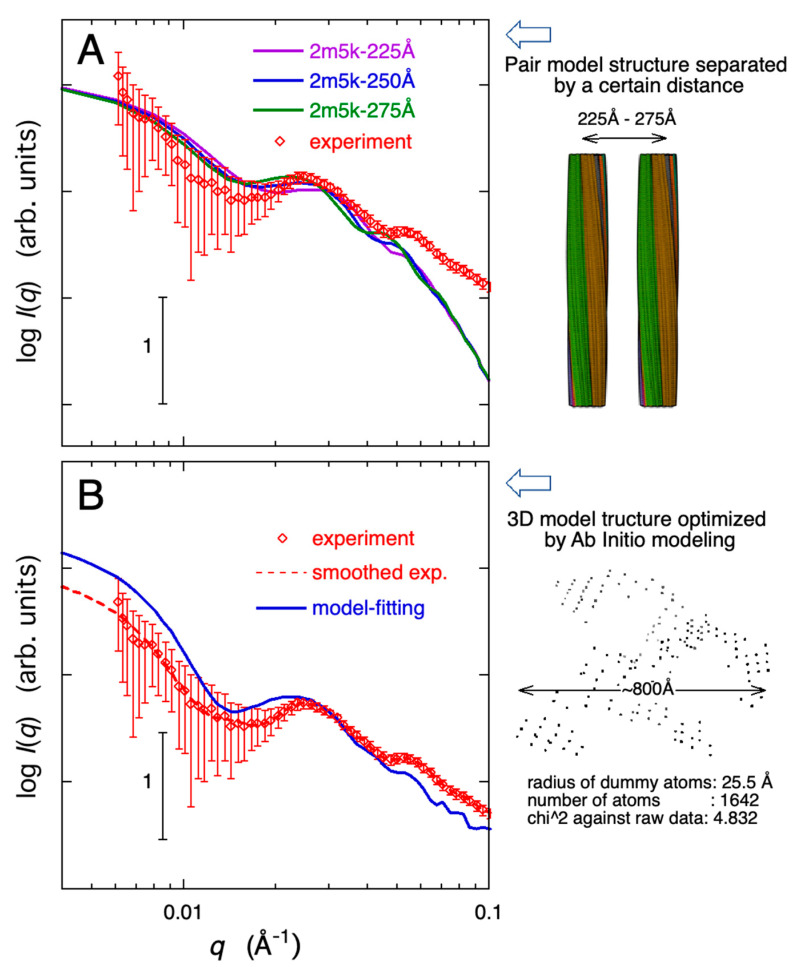
Comparison of the experimental data with the theoretical scattering curves of higher-ordered fibril models. The experimental scattering curve of the Aβ protein, observed after 32.6 h in Figure 6B, is overlaid. In (**panel A**), the theoretical scattering curves are derived from pairs of 3D models of 2m5k (depicted in Figure 9) arranged in parallel at fixed distances (225 Å, 250 Å, 275 Å). In (**panel B**), the theoretical scattering curve is optimized using the ab initio (beads) modeling method with the DAMMIF program. In both modeling approaches, the observed correlation peaks at approximately 0.025 Å^−1^ and 0.05 Å^−1^ are qualitatively reproduced.

**Figure 11 molecules-28-06555-f011:**
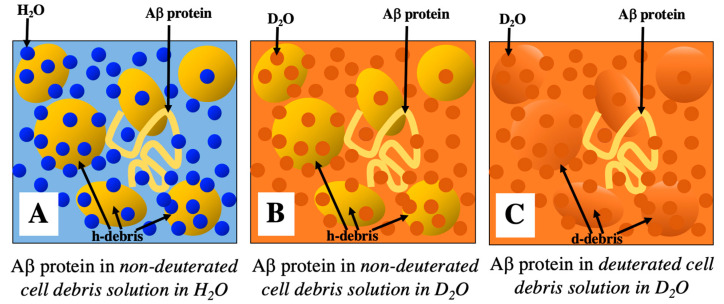
Conceptual diagram of a novel experimental method using neutron scattering for the selective observation of the structure of a specific molecule of interest in an intracellular molecular crowding environment. The string represents the target protein (Aβ protein), while the spheroids depict crowder molecules (intracellular components). (**Panel A**,**B**) show scenarios of neutron scattering measurements using non-deuterated compounds dissolved in H_2_O solvent and D_2_O solvent, respectively. (**Panel A**) is the equivalent scenario for X-ray scattering and conventional spectroscopy. In both (**panels A**,**B**), both target molecules and a significant amount of crowders are detected. (**Panel C**) depicts neutron scattering utilizing deuterated cell debris, with the average scattering density of the debris matched to that of D_2_O. In this instance, the structure of the target molecules is predominantly detected.

## Data Availability

The data presented in the manuscript are available upon request following its publication (mhirai@gunma-u.ac.jp).

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
