# Peer review of "Fibrillization Process of Human Amyloid-Beta Protein (1–40) under a Molecular Crowding Environment Mimicking the Interior of Living Cells Using Cell Debris"

_molecules, 2023, doi:10.3390/molecules28186555_

Round 1

Reviewer 1 Report

The authors present a report about fibrillization process of human amyloid-beta protein under a molecular crowding environment. Cell debris solutions with different D2O/H2O ratios were examined by using neutron scattering. Then Aβ protein dissolved in D2O solvent with and without the deuterated-cell debris solution were studied. Based up the obtained neutron scattering curves, structural models of amyloid fibrils were constructed. Subsequent analyses suggest that intracellular environment accelerates the fibrillization of Aβ proteins and the formation of higher-ordered aggregates. Overall, this manuscript is well organized and interesting. I think the manuscript is suitable for publication after revisions.

1. The time evolution of the scattering curves of deuterated-cell debris solution without Aβ protein should also be provided as a control.

2. Limited information was provided in Materials and Methods. More detail about the inverse contrast matching technique and model construction should be included.

Reviewer 2 Report

In the present study the authors investigated the fibrillization process of human Aβ1-40 under molecular crowding environment by using neutron solution scattering technique. Neutron scattering technique possess several advantages as compared to traditional X-ray, for example no radiation damage to solute molecules caused by free radicals. In addition, the authors were able to demonstrate the effectiveness of deuterating the cells to reduce background scattering from cell debris, and this environment facilitated the maturation of higher-order Aβ1-40 fibrils in vitro.

The study is interesting and important. The manuscript is well written, and I only have minor comments:

1. Page 12, line 502: Please add references.

2. Page 12, lines 509 and 514: Please use a Greek symbol for Ab.

3. Can this method be used in APP-transgenic mice, a mouse model of Alzheimer disease?  This should be stated in the discussion.
